# Botched Ebola Vaccine Trials in Ghana: An Analysis of Discourses in the Media

**DOI:** 10.3390/vaccines9020177

**Published:** 2021-02-19

**Authors:** Esi E. Thompson

**Affiliations:** Communication Science Unit, The Media School, Indiana University Bloomington, Bloomington, IN 47405, USA; Esiethom@iu.edu; Tel.: +1-812-855-1726

**Keywords:** moral panic, risk society, Ebola vaccine trials, Ghana

## Abstract

In June 2015, proposed Ebola vaccine trials were suspended by the Ministry of Health of Ghana amid protests from members of parliament and the general public. Scholarship has often focused on the design, development, and administration of vaccines. Of equal importance are the social issues surrounding challenges with vaccine trials and their implementation. The purpose of this study was to analyze discourses in the media that led to the suspension of the 2015 Ebola vaccine trials in Ghana. I use a sociological lens drawing on moral panic and risk society theories. The study qualitatively analyzed discourses in 18 semi-structured interviews with media workers, selected online publications, and user comments about the Ebola vaccine trials. The findings show that discourses surrounding the Ebola vaccine trials drew on cultural, biomedical, historical, and even contextual knowledge and circumstances to concretize risk discourses and garner support for their positions. Historical, political, and cultural underpinnings have a strong influence on biomedical practices and how they are (not) accepted. This study highlights the complexity and challenges of undertaking much needed vaccine tests in societies where the notion of drug trials has underlying historical and sociological baggage that determine whether (or not) the trials proceed.

## 1. Introduction 

The quick spread of the 2014 Ebola outbreak in some West African countries reignited vaccine development for new and emerging infectious diseases as an issue of priority. Vaccine controversies have received some attention in scholarship. Recently, the focus has been on vaccine hesitancy, a situation in which individuals refuse or are unwilling to accept vaccines [1]. Research has also shown that vaccine trials can be met with opposition and has suggested various measures for addressing this [2,3,4]. Other scholars have focused on the atypical situation of vaccine trials in developing contexts. Lindegger, Quayle, and Ndoluv, for instance, studied communities’ preparedness for HIV vaccine trials in South Africa and found that although there were positive attitudes about vaccines in general, there were anxieties and concerns about the safety of the HIV vaccine trials [5]. 

Irrespective of vaccine hesitancy or vaccine trial controversies, the media play a role in providing information for individuals to create their views and attitudes. Bearing in mind the key role that the legacy media in developing contexts play in health communication and public health campaigns [6,7,8], understanding discourses about vaccines in the media merits research. 

Scholarship has often focused on the design, development, and administration of various vaccines. Of equal importance are the social issues surrounding vaccine trials and their implementation. Recent push-back against suggestions for testing coronavirus vaccines in Africa brings up often neglected historical, colonial, and social concerns surrounding responses to vaccine trials (e.g., [9,10]). Such situations call into question how antagonistic responses to vaccine trials are often viewed as resistance or misconceptions without considering the socio-historical and contextual issues that precipitate the responses. 

The current study uses the halted Ebola vaccine trials in Ghana as a case study to unearth political, social, and historical factors that led to the halting of the trials. Two trials: Phase 2 clinical trial of the Glaxo Smith Kline candidate vaccine and phase 1 of the Johnson and Johnson clinical trial [11]—were undergoing review for implementation in Ghana. The Food and Drug Administration (FDA), had given conditional approval to Janssen, a subsidiary of Johnson and Johnson Pharmaceuticals, for preparation to conduct phase 1 trials for their candidate vaccine in Hohoe in the Volta region. This became the issue of contention, and eventually all proposed Ebola vaccine trials were suspended. 

Using analysis of semi-structured interviews with media workers, selected online publications, and user comments, the study explores discourses at work in the media in the suspension of the Ebola vaccine trials. This research is relevant now, as emerging infectious diseases such as Zika, Middle East Respiratory Syndrome (MERS), and the recent Coronavirus suggest increased need for vaccines. Again, how various platforms are used to support or impede vaccination development and dissemination efforts in developing contexts should be of interest to scholarship in general. Lastly, communication about vaccines has tended to focus on promoting the uptake of vaccines, but resistance to vaccine trials suggests that research should cover the entire lifecycle of vaccines from development through trials to roll out. 

Rather than viewing push-back against vaccine trials as ignorance, the study argues that the historical and politico-cultural and economic contexts within which such views are expressed need examination. Specific to the current study, rather than lay people express resistance, experts and politicians equally express resistance, further muddling the idea that expressions of resistance are borne out of ignorance and misinformation. Adopting a sociological lens in analysis helps to unpack and uncover underlying factors that frame the resistant reaction, providing insights not immediately obvious. 

### 1.1. The Issue—How It Evolved

On 21 May 2015, a local radio station published a story on its digital platform, starfmonline.com (accessed on 3 June 2015), about a planned Ebola vaccine trial at the Hohoe Midwifery Training College in the Volta region of Ghana. The story suggested that officials of the University of Health and Allied Sciences in Ho were leading the trial. An anonymous source indicated that the candidate vaccines had been successful in animal tests and were ready for human testing. The volunteers were to be compensated with GHC 200 (equivalent to $50 at the time), and a mobile phone, among other incentives. The story quoted some students who indicated being scared and not knowing the risks involved in the trials. A source at the FDA confirmed the trials but did not confirm the approval stage of the trials. The story also cited the lead investigator and Pro-Vice Chancellor of the University of Health and Allied Sciences, Prof. Binka, who confirmed the trials as part of preparation towards Ebola case detection in Ghana.

The next time the issue came up was when the Coalition for Ghana’s Independence Now (CGIN), a group in the Volta region, issued a press release on 30 May 2015, condemning the vaccine trials and calling for an end to them. The press release suggested that the “criminal” trials will spread Ebola in Ghana because healthy people will have to be infected with the Ebola virus before being given the vaccine. 

The Volta regional branch of the National Democratic Congress (NDC; the party in government at the time) published a press release calling for an end to the trials a week later. Their press release described the vaccine trials as “unfortunate,” risking the lives of Ghanaians as guinea pigs for an “unnecessary experiment.” It was at this point that the media picked up the story and published interviews with stakeholders about the Ebola vaccine trials. The FDA which has the mandate to regulate, approve, monitor and assess clinical trials in Ghana, held meetings to discuss the situation and subsequently approved the Hohoe trials (which hitherto had not been approved). There was an emotive discussion on the floor of parliament (highest law-making body in Ghana) following this decision. On 10 June 2015, the Minister of Health suspended all Ebola trials. 

### 1.2. Ghanaian Media

The media in Ghana have grown and expanded from the pre-fourth republic days of censorship and suppression to a free, vibrant, and thriving media operating in a democratic environment with guaranteed freedoms [8,12]. In 2015 when the trials were suspended, there were 354 operational FM radio stations, 34 television stations, and about 25 newspapers (www.nca.org.gh). Most Ghanaians depend on FM radio stations as trusted and credible media that broadcast in various local languages, thus eliminating the literacy barrier [8,13]. Newspapers are well respected because of their agenda setting role through newspaper review programs on FM radio stations [14,15], but are limited because they publish in English in a country with high illiteracy [13].

The media are embedded in societies and have the symbolic power to shape events, outcomes, individual and social perceptions, beliefs, and actions [16]. How media workers construct and use this influence is through ideologies (individual journalists and media organizations), professional practices, and wider social debates about ongoing issues. The struggle over the vaccine trials occurred within the media because various stakeholders used the media to present their messages and positions. However, as media sociologists remind us, the media do not just publish information; choices are made regarding the type of information to publish, how to frame the information, the words to use, as well as selection of sources [8]. Therefore, understanding discourses in the media about the Ebola vaccine trials unearths the various contested positions, suppositions, and contextual factors that vaccine trials need to consider. 

## 2. Literature Review

The study draws on literature on moral panics [17] and risk society [18,19] to analyze how the vaccine trial issue played out in the media. Both theories focus on sites of social anxiety and fear. 

### 2.1. Moral Panic 

Moral panic, developed by Cohen [17] and expanded by others [20,21], explains societal reactions to threatening conditions. Drawing from sociological theory, Cohen addresses collective behavioral and social processes in situations where a society believes its existence, well-being or interests is threatened [17]. In such situations, a “scapegoat” is constructed and disseminated through the media as the cause of the situation. These “folk devils” or “moral deviants” are stigmatized and used to justify social actions or controls [22]. Through a process of claims making by ‘moral entrepreneurs (experts, politicians, etc.), judgment is made about the issue and the deviants to protect social values from moral threats [16]. Moral panic sees the media as a main actor in the spread of the panic/evil either consciously or unconsciously. In the original words of Cohen, periods of moral panic occur when: 


*“A condition, episode, person or group of persons emerges to become defined as a threat to societal values and interests; its nature is presented in a stylized and stereotypical fashion by the mass media; moral barricades are manned by editors, bishops, politicians and other right-thinking people; socially accredited experts pronounce their diagnoses and solutions; ways of coping are evolved or (more often) resorted to; the condition disappears, submerges or deteriorates and becomes more visible”*
[17] (p. 9) (emphasis added).

Good and Ben-Yehuda outline five conditions that a situation must satisfy to achieve the status of a moral panic [21]. They suggest that the issue or condition must be *volatile* even if it is one that has been in existence. The condition or the people attributed to the condition must also be viewed as contradicting the values of society and thus be strongly resented by the society. Third, the threat must be viewed as one that can be concretely determined or quantified. Fourth, the condition or issue must cause or be viewed with deep concern among a substantial part of the population who view the threat or condition as real and threatening. Finally, the threat must be viewed as much more serious than can be proven or verified. Moral panic can have varying levels of intensity and effect, may be related to trivial, serious, or even nonexistent issues, can have long lasting or short-lived impacts, and can be spontaneous or simmering [16,23]. 

Moral panic has been generative and thus critiqued for losing its original meaning with confusion about what counts as a moral panic. Others have critiqued its implied powerful media and single passive audience [24]. In response to these critiques, and with a view to returning to the original focus of moral panic, scholars are moving from a focus on *panic* to a focus on *moral boundaries,* which emphasizes the *perception* of a threat and not the magnitude of the threat [23,25]. 

Other scholars view the changes in society as necessitating broadening the conceptualization of moral panic to accommodate these changes [16,26]. This latter approach better fits Cohens original definition of “…condition, episode, person or group of persons…” by including new sites of anxiety, rather than the narrow focus on only ‘stigmatized groups’ and also allows for more actors as moral regulators [16]. The downside of such an approach is the possibility of moral panic becoming a catch-all concept and suffering from meaning drift [16,27]. I adopt the latter view of moral panic, but hold to conceptual soundness by focusing on discourses in the media about the issue. Here, the focus is not on the scare but how it was constructed in relation to other social anxieties. 

### 2.2. Risk Society

For Beck, risk society addresses the risks and hazards introduced into society due to modernization, technology, and development [18]. Beck argues that although early modernization was characterized by positive man-made outputs (increased incomes, technologies, etc.), the associated new negative harmful effects and risks of modernization are being felt in late modernization, (e.g., industrial disasters, chemical leaks, air pollution, etc.). These negative effects raise concerns about future safety that need to be addressed in the present, leading to a reflexive orientation associated with increased scrutiny and public criticism of scientific and technological developments. Beck suggests that in many societies, there are “institutions of monitoring and protection” whose duties are to protect the society from “social, political, economic and individual risks” [28] (p. 303). Here too, the media play a vital role as the platform within which the risks are defined, made known, and the (in)actions of various actors exposed for public scrutiny [29]. 

Risk society theory has been critiqued for assuming a single monolithic media contrary to the pluralistic and fragmented media projected in media studies. 

While moral panic focuses on the morality and value systems threatened in the sites of anxiety, risk society focuses on the irreversible potential catastrophes associated with technological or industrial development within those sites of anxiety. Again, unlike moral panic which has distinguishable folk devils, the identification of deviants in risk society is a “foraging process” to find liable parties [30] (p. 281). For Durodier, the focus should not be on whether moral panic or risk society has increased in their seriousness, but rather that people *believe* this is so and act accordingly [31]. Similarly, Beland argues that while some scares take the form of a sensational, exaggerated moralizing consistent with moral panic, others are knowledge-based discourses about anxieties and risks associated with risk society [32]. Beland thus argues for a case-by-case empirical investigation that could explain the scare. This study focuses on the suspension of the Ebola vaccine trials in Ghana as a case of social anxiety. 

## 3. Method

I adopted a qualitative approach in this study. Specifically, I used semi-structured in-depth interviews with18 journalists in legacy media (television, radio, and newspaper) in the capital city of Ghana and qualitative content analysis of online media publications and user comments. The choice of online media was based on the fact that research has shown that both television and radio media publish similar or more content on their digital platforms than they broadcast [33]. 

Although radio is the most preferred legacy media in Ghana, radio broadcasts are not well archived. I used the online publications of the leading radio stations in Ghana instead. These are limited in their reach due to accessibility and literacy issues but they provide a diverse enough sample because unlike newspapers, online stories can be updated anytime with new information. 

Furthermore, the platforms allow user comments, which provide a form of user insights for analysis. Data for the study were obtained from three sources: interviews with media workers who provided a form of metacommunication about media reports on the vaccine trials, online news media content, and user comments on these online publications. The study formed part of a larger study on risk perceptions about Ebola in West Africa. 

### 3.1. In-Depth Interviews

Qualitative interviews were carried out with a purposive sample of media workers in Accra, the capital city of Ghana, which hosts the majority of news organizations in the country. For individuals to be part of the study, they had to have worked with a legacy media organization during the Ebola crisis as news gatherers, anchors, producers, presenters, or editors and worked on stories about the Ebola vaccine trial. Two approaches were used in recruiting: First, invitations were sent to all registered legacy media organizations in August 2015. I followed-up with phone calls to confirm participation. I also purposively contacted editors and journalists in legacy media that had published stories about the Ebola vaccine trials. Through snowballing, I interviewed other participants that were recommended. In total, 18 legacy media workers participated in the study. Data gathering occurred between 22 August and 18 September 2015, and was done in English. All interviews except for two were conducted in the media house. Interviews lasted between 30 and 60 min. When no new information was received and recommendations were of media workers who had been interviewed, data saturation was deemed to be achieved. 

#### Instrument 

A semi-structured interview guide was used in undertaking the interviews. Four main questions along with probes and prompts made up the interview guide. The first question addressed what respondents focused on when reporting about the Ebola outbreak in general. The next question addressed respondent’s familiarity and choices regarding the Ebola vaccine trials (when they heard about the issue, through whom, what they did with that information, choices of sources used in stories, choices of press releases to publish, etc.) The third question addressed respondent’s perception about the impacts of the media reportage about the Ebola vaccine issue, while the fourth question addressed what respondents would do differently/similarly with future situations. The interview guide was pilot tested with other journalists prior to the study. Each respondent provided written consent to participate in the study. I guided the interviewees by the questioning, while the interviewees led the interview through the stories, interpretations, and responses they provided. 

### 3.2. Content Analysis

I purposively sampled the digital news sites of four radio stations (Myjoyonline.com, citifmonline.com, peacefmonline.com, and starrfmonline.com), the leading newspaper (graphic.com.gh), and a well-known digital news site (ghanaweb.com). Citifmonline.com and myjoyonline.com are the online platforms of the leading English language radio stations, while Peacefmonline.com is the digital platform for the leading local language radio station. Graphic.com.gh is the online platform of Daily graphic, the widest circulating state-owned newspaper in Ghana, while starrfmonline.com is the digital platform that broke the story. Stories had to have been published between 20 May and 15 June 2015. News stories, features, and editorials published about the Ebola vaccine trials were downloaded and saved in pdf for analysis. I omitted advertisements and stories tangentially related to the Ebola vaccine trials.

I also analyzed user comments on these stories. Commenting is a way for users to engage with a story and in public discussion of events. For the media, user comments provide a way of engaging community and promoting loyalty [34]. The majority of users read comments even if a very small proportion actually post comments [35]. Thus, user comments, which provide context for the story, can influence the perceptions of readers [36]. In total, 21 stories and 245 user comments were content analyzed. I do not provide the names or handles of users to protect their privacy because users wrote their comments without expecting that they will be cited. 

### 3.3. Coding and Analysis

Interview recordings were transcribed verbatim for analysis. I used three-step coding in analyzing the data [37]. First, initial or open codes were developed based on a reading of the data. Here, emic words or words used by the interviewees or in the stories were used to break down the data. Next, I used axial coding to connect, create categories, and build relationships among the initial codes. Finally, at the selective coding stage, I collapsed categories under larger themes. Appendix A shows the sample coding and analysis process. 

The findings from the interviews are presented along with the results of the content analysis to show the discourses employed in constructing (or deconstructing) the issues leading to the suspension of the trials.

## 4. Findings

Media workers presented ambivalent positions about coverage of the vaccine trials. While some expressed support for how the media covered the issue and unveiled evil cover-up, others condemned the media for poor journalism and tarnishing the image of journalists. 


*“We [media] weren’t told that this thing was going on. Parliamentarians didn’t even know about it until we raised the red flag. When parliament saw how the situation was going, they had to call in the minister who was then out of the country to come and explain what was going on. There were so many questions as to how it was going to happen.”*
(Local language radio journalist).


*“I was coming to work and I was listening to the radio and the presenter was so confident of the questions he was asking. He was looking at the WHO website and asking a scientist a question about trials in Ghana. He was rather disgracing himself. The scientist was saying the trials had not started and the presenter was saying they had.”*
(English language television anchor and broadcast journalist).

Other journalists chastised the initial story in hindsight but admitted that they would have made similar choices. They based this decision on the nature and competition in journalism practice in Ghana which makes journalists eager to break a story. 


*“In Ghana, there is competition in the media. If I get a story that they are going to have an Ebola test…; Ebola, a dreaded disease that everybody was praying never comes to Ghana and so far, we never recorded any case; and an organization is coming to have the vaccine test; with the background knowledge that some vaccine tests have gone bad some time past, I won’t hesitate to write that story.”*
(Private newspaper journalist).

Other interviewees suggested that the suspension of the vaccine trials was because the issue was politicized. Both majority (NDC) and minority (National Patriotic Party: NPP) political parties suggested that the vaccine trials were not in the interest of their constituents or Ghanaians. This discourse was evident also in the online publications. The proposed trial site, the Volta region, is the stronghold of the ruling NDC party and also hosts one of two international research centers. The minority party presented the vaccine trials as a failing of the president for allowing the trials as a public service radio and television journalist alluded to: “The NPP said, ‘no you can’t use Ghanaians for the test trials.” A senior television journalist described the position of the NDC thus: 


*“The regional minister politicized the whole issue in a way that he was trying to say he was protecting his people: ‘I have noticed the issue; Ebola is this deadly disease so let me in a way bring out the notion that I am protecting my people.”*


Headlines such as “Mahama should take the Ebola vaccine trials to Bole, V/R NPP chairman,” published on myjoyonline.com on 11 June 2020, and “Ebola vaccine trials-Volta NDC questions compensation package,” published on Citifmonline.com on 14 June 2020, reflect these sentiments. In the former example, the NPP Volta Regional chairman accused the president of complicity and suggested that the trials should be held in the home region of the president if indeed the trials were safe, while in the latter story, the Volta regional NDC office questioned the puny compensation package for the volunteers and why the trials were kept under cover. Users posted similar comments online. In all these cases, the vaccine trials became a trump card to promote a discourse of public interest and concern and to ramp up support for each political party. 

An interviewee suggested that vaccine trials, for some volunteers, is a livelihood and a source of income. 


*“… they did not have in mind that there have been so many vaccine trials in Ho and the people were willing to do it because vaccine trials were a form of livelihood for some of the people in the community. Some said they had done it many times and for them it was their job.”*


The radio news anchor was suggesting that rather than agentless, fearful volunteers, the volunteers had agency and were aware of what the trials entailed or at least were willing volunteers. For other interviewees, the vaccine trials allowed constitutional rights to be ignored. 


*“We wouldn’t have toiled and developed a vaccine and then when we say we are going to test the vaccine, you say we should retract [the story] because we spoke to an expert who expressed worry about the confusion the parliamentarians were sending about the vaccine.”*
(Television news editor).

Such media workers were concerned that the very constitutional basis of their profession—freedom of the press—was trampled upon because they had allowed views that others found offensive. This argument was probably in relation to the TV3 network which was asked to retract a story in which Prof Dodoo, former chair of the World Health Organization’s Collaborating Centre for Pharmaco-Vigilance, had accused parliamentarians of ignorance about vaccine trials and the FDA. Many stories in the media included views from both trial opposers and trial supporters likely in an effort to present both sides of the issue. 

In the online publications, some Members of Parliament cited a lack of communication and public sensitization for such a trial as the basis for their opposition. This claim implies the trial investigators were ignorant about cultural and social norms. Some interviewees suggested that how other media workers covered the issue was also based on ignorance: 


*“The journalist that wrote that story did not take the time to go and research what a vaccine is, what a vaccine test entails, what an Ebola vaccine test will do to the volunteers… that is basic journalism,”*
said a public service newspaper journalist.

Some media workers made claims of uncovering a potentially deadly trial. However, as the extract below suggests, it was less the vaccine trial itself and more of the perception of Ebola that drove how the media handled the issue: 


*“What triggered the whole thing was the sensitive environment. We have never had a confirmed case [of Ebola] in Ghana; it has always been suspected so even if you are doing some trials, at least let us know how it is going to go.”*
(Public radio presenter and journalist).

Interviewees and user comments constantly referred to Ebola’s virulence and the fact that God had protected Ghana from infections to explain why the trials were dangerous for Ghanaians. Some user comments and stories in the online publications suggested that the vaccine trials would lead to Ebola infections in Ghana. This position was given intellectual weightage by the Ghana Academy of Arts and Sciences (GAAS), a reputable and prestigious group of eminent senior scholars in the arts and sciences. The Academy released a press statement, which, among other things, sought to find out the composition of the candidate vaccines, the type of adenovirus in the trial sites, and assurance that possible mutation was not a risk before the trials are approved. The story, published on graphiconline.com.gh was headlined, “Its unsafe to undertake Ebola vaccine trials.” The perception that scientific experts were divided over the vaccine trials raised more fears and gave further credence to the position of the trial opposers. 

Users commented most on the initial online story with the vast majority condemning the trials. Many commenters raised issues about safety, the greed of leaders who were willing to sacrifice their citizens for personal gain, the gullibility of people who would sacrifice themselves for pittance, that the trials would lead to Ebola cases in Ghana, or related the trials with the infamous Tuskegee syphilis trials. Other users suggested that the vaccine trials were population reduction attempts by Westerners. Users posted similar comments when the FDA approved the trials. The extract below illustrates this:


*User 5: “Tests can be done but those to be tested should not be lured with material things. People are used for tests always, especially Blacks. Do you remember the Alabama syphilis tests on Black soldiers? Ghanaians needs not start such a test on a particular group of people.”*



*User 45: “Big Western Pharmaceuticals want to use Ghanaians and Africans as cheap TEST-PIGS! They don’t care. It’s Ghanaians and Africans who die! All over Europe, and USA - a “Test-Pig” would be paid, at least Euro 6000! Including full hospital stay, and all medical tests! Advise: Save your health! Save your life! Refuse any-kind of ‘Test Vaccine’”*



*User 88: “This vaccine is very new and so how can they be sure that it’s safe and people won’t end up developing the Ebola virus?”*



*User 4: “And how can you give yourself out for GHC 200 and a phone? All in the name of Ebola vaccination and the so-called other things.”*



*User 183: “Hey my people don’t accept such vaccines. Evil minded people ruling us. They just wanted to reduce the population so that the burdens of the country will come down a bit to create, loot, and share our resources. It will never happen. Be wise my people from Volta.”*


These comments suggest the diverse interpretations that users have about what a vaccine trial is and how it works. Secondly, it reflects deep seated historical parallels about trials and the value of life that users draw on to interpret and make sense of the vaccine trials. It also shows that the public uproar did not result from the positions taken by the members of parliament, the CGIN, the volta regional NDC or even by how the media covered the issue. Rather, there exist social and historical frames of reference common to both the regulators and ordinary Ghanaians which influenced responses and reactions to the Ebola vaccine trials. This frame of reference draws on historical trials such as the Tuskegee syphilis trials on African American males and the value placed on the life of trial volunteers. 

In the user comments, there were long threads arguing for why the trials were not needed and should not be allowed to take place. Words such as “unnecessary”, “needless”, “criminal”, and “unfortunate” were used to describe the trials while the volunteers were described as “guinea pigs” and “scared.” Other metaphors used to describe the trials in the online publications and user comments were “an ambush of Ghana”, the trial investigators had “sold their soul”, “secret Ebola vaccine”, “cruel Ebola vaccine trial”, etc., all suggesting clandestine activities. 

Although many parents vaccinate their children against the five childhood killer diseases in Ghana (The WHO-UNICEF estimates 90% childhood vaccination rate in Ghana in 2015 (https://data.unicef.org/wp-content/uploads/country_profiles/Ghana/Immunization_gha.pdf (accessed on 24 December 2020)), trial opponents argued that the trials should be undertaken in countries that have recorded Ebola cases. One user commented, “Are they Ebola patients? If not, why should you try the vaccine on them.” Another wrote:


*“They should do it in countries that were affected with Ebola. Why Ghana? Did we have such a problem? Are we experiencing it? I totally do not want this to happen because Ghana cannot be a target population for a trial of a disease that did not affect us.”*


Trial investigators and other experts appealed to science and social benefit by making the following claims: the trials would project Ghana in a positive light; the vaccines have been tested in other places previously and are safe; the FDA had expertise to review the trials in the interest of Ghanaians and have been approving trials in the past; and that trials are beneficial to science and the academy. They also tried to debunk the notion that the trials had started. In a story headlined “FDA approves Ebola vaccine trial in Ghana,” published on 10 June 2020, the public affairs director of the FDA is quoted as saying, “The most important thing is to establish whether the vaccine is safe and that has been established. It’s also established that the vaccine is not going to cause any Ebola disease.” User comments on this story reflected disbelief in these statements and antagonism towards the FDA for approving the trials.

News stories and user comments framed the media and opposers to the trials as saviors while the trial investigators, pharmaceutical company, and the FDA were presented as traitors who were willing to sell their people for money. The latter were often captured as conniving with evil foreigners to infect Ghanaians with the Ebola virus and hiding the true nature and purpose of the trials. One user wrote:


*“Prof. Binka and Prof. Koram are both Epidemiologist and not Virologist but they want to carry on with an Ebola clinical trial in Ghana. These are just selfish physicians wanting to use fellow Ghanaians as Guinea pigs for their personal interests. It is time the government steps in and stops this CRIME.”*


This was buttressed by constant reference to the compensation package which was viewed as incommensurate with the risk involved in participating in Ebola trials. 

## 5. Discussion

I investigated discourses surrounding the suspended Ebola vaccine trials in Ghana from the media. Below, I discuss the risk society and moral panic discourses that emerged as well as the main themes. 

### 5.1. Moral Panic Discourses

Moral panic discourses of stigmatizing, expert credence, and folk devils were all evident in the data examined. Stigmatizing discourses drew on claims that the vaccine trials were being planned in secret; doubts about the safety of the trials and the possibility of the vaccine causing Ebola virus disease; that the trial implementers were selfishly selling their citizens; and that the compensation package was not commensurate with the risk involved. The supposed volunteers were also stigmatized for offering themselves for pittance. These claims were given further credence by the GAAS press release. GAAS’ position resonated with the public because the questions reflected deep-seated anxieties and public fears about the Ebola virus disease. Convergence of public and expert concerns legitimized the idea that the trials would lead to Ebola virus infections in Ghana. The claims evoked the idea that a group of people was breaking the moral and ethical responsibility of protecting the lives of Ghanaians. 

Different groups of moral entrepreneurs arose around the issue. The first moral entrepreneur was the media that was viewed as unveiling evil happenings in the society. Investigative journalism has in recent times exposed corruption and evil dealings in Ghana [38]. Therefore, the media’s claim of uncovering and exposing this “clandestine activity” was in fulfillment of a moral agenda to protect human lives and livelihoods. The most respected moral entrepreneur was the GAAS, who issued a statement recommending pausing trial approvals until all relevant questions had been answered. Coming from a well-recognized expert scientific group, GAAS’ claims had both moral and expert value and were regarded as authoritative, legitimate, and accurate. The final group of moral entrepreneurs was the members of parliament. This group of policy makers provided the last barricade against the trials. The discussion of the issue on the floor of parliament, not only gave weight and importance to the issue, but showed that the threat was real and needed to be addressed. 

The trial investigators, the pharmaceutical company that produced the trial vaccines, and the FDA who approved the trials were all viewed as folk devils. Claims that these groups were breaking moral laws and selling their people as guinea pigs in a needless experiment were rampant in the media. They were condemned, vilified, presented as not interested in the health and welfare of Ghanaians, placing personal interest above the health of people, and trying to decimate the population.

### 5.2. Risk Society Discourses

The risk society context predisposes people to view things from a skeptical standpoint and provides a way of dealing with unknown risks. Some news stories, user comments, and media workers constructed the Ebola vaccines as a manufactured risk with unknown effects that had the potential to threaten lives and health. The Ebola vaccine was viewed as carrying the Ebola virus or requiring that one be infected with the Ebola virus before being vaccinated. 

Second, the discourse of safety of the vaccines was raised by both experts and lay people to justify suspension of the vaccine trials. The anxiety associated with safety was borne out of an existing fear of the Ebola virus disease itself [8], questions about its origins, and why Ghana, which had recorded no cases, should be included in trials. 

The risk society context also creates an environment where trust in science is low and there is more questioning of scientific ventures based on historical knowledge. There was an obvious lack of trust in science with scientists viewed as working in their personal interest (although GAAS’ position was supported because it aligned with that of the lay public). When push-back against the trials did not yield the expected response, policy makers stepped in to stop the trials. When science seems not to be trusted, politics takes precedence and paves the way for policy and action. 

Although Ghana has not had previous issues with vaccination side effects or a direct history of trials gone wrong, the data showed that various stakeholders drew on knowledge from different contexts both far and wide to guide their responses. This suggest that a risk, whether real or perceived, does not need to be experienced to become part of one’s frame of reference. The very idea of a risk society creates a wealth of historical information that respondents can tap into to make sense of their current circumstance. 

### 5.3. When Moral Panic and Risk Society Converge 

Moral panic can lead to alternative knowledge claims that challenge existing knowledge and provide a critical or alternative lens for viewing risks. Such alternative medical interpretations may lead to policy that could be ineffective or misguided at best and punitive at worst. The trial investigators and their supporters viewed the trial as less risky, focused on current science, and excluded moral questions and cultural expectations, but the popular view held that the trials were very risky and had the potential of spreading Ebola based on historical knowledge of previous experiments. They also viewed the trials as immoral and unethical. 

Here, there is a clash in biomedical knowledge claims and popular or alternative knowledge claims. Explanations of safety from experts leading the trial did not seem to make any difference, as Ungar has noted that “a safety model … is not readily sold to a public whose demands for a yes/no risk evaluation hardly countenances a cost-benefit analysis” [30]. The alternative explanations recall and reinterpret biomedical and historical information and modify and appropriate them to make sense of current happenings [39]. These alternative explanations are often viewed as misconceptions or ignorance and in so doing privilege biomedical explanations while dismissing alternative explanations. But, the suspension of the Ebola vaccine trials suggests the need to harness both biomedical and alternative knowledge claims without privileging one over the other. It also requires working through existing cultural and social systems and structures to reach understanding and consensus in a pluralistic and respectful way.

Constant reference to the Tuskegee syphilis trials and the polio vaccine trials in Nigeria [40], to buttress the position of the trial opposers suggests that historical antecedents are the context for current happenings and may lead to ignoring new information. It is harder to change incorrect information after it has spread. Therefore, seeking consensus and leading the discourse will help to address misinformation. 

Parliamentarians claiming concerned and scared electorates as the basis for their opposition built an affective relation between themselves and their constituents. They thus ‘gave voice’ to the people’s concerns by speaking in the interest of “the people.” By so doing, “the people” are constructed as fearful, voiceless, and victims of the greed of others. Such constructions helps to hold the authorities (FDA, Prof Binka, etc.) responsible and present them as complicit deviants or at least enabling the deviants to operate. The faceless citizens are presented as agentless with their agency expressed through the parliamentarians and other civil society organizations. However, user comments suggest that users have agency and express this agency through their posts. Furthermore, volunteers arguing that participation in trials is a livelihood also projects the agency that “the people” have. 

The idea of “public interest” was coopted by various stakeholders to support and justify their positions and declarations. For the FDA and the trial implementers, it was in the public interest to participate in an international vaccine trial that would benefit humanity and which would enhance Ghana’s reputation in the scientific and academic fields. For the CGIN and the NDC, it was in the interest of their constituents to speak up against the trials and call for their suspension. For the MPs, it was in the interest of Ghanaians for them to speak up and defend human lives from unknown harm. For journalists, it was in the public interest to investigate and reveal clandestine activities that threaten public health and wellness and to push political leaders to act on behalf of the people. Public interest then becomes a crutch that each stakeholder depends upon to distinguish and elevate their motives from any assumed or perceived personal or professional gains. 

### 5.4. Concluding Remarks

This study highlights the complexity and challenges of undertaking much needed vaccine tests in societies, where the notion of trials has underlying historical and sociological baggage. I argue that the discourses surrounding the Ebola vaccine trials draw on cultural, biomedical, historical, risk perception, and even contextual knowledge and circumstances to concretize risk discourses and garner support for positions irrespective of the veracity of the claims made. Historical, political, and cultural underpinnings have a strong influence on biomedical practices and how they are (not) accepted.

The vaccine controversy reflects a performance that intertwined individual and political/professional agendas in the context of the existing fear of the Ebola virus. The impacts of these actions were delayed testing of a much-needed vaccine and Ghana’s potential challenges with accessing such vaccines in the hopefully unlikely case of an Ebola outbreak. 

However, at the same time, it shows the need for more open communication and engagement on vaccine issues in contexts where historical, colonial, and present realities of mistrust intersect with fears about how such activities are interpreted. This paper also unearths knowledge gaps that exist in relation to Ebola as a disease and the associated biomedical interventions (in this case vaccine trials) geared at addressing them. More broadly, it shows limited understanding about vaccines and how they work, while highlighting the fears that are associated with vaccines (not unrelated to the current anti-vaccine scare in developed countries) [39]. 

It also reflects the low knowledge of science and health communication in general among the media and the need to engage the media generally on health and science issues. Journalists in Ghana do not specialize, and science and health reporting are second to politics and the economy [8]. This may explain the limited knowledge that media workers had and how they handled the Ebola vaccine trial controversy. 

The findings of this study should be interpreted within certain boundaries. First, this qualitative study used purposive and snowball sampling which do not allow the findings to be generalized. Secondly, the media workers sampled were all in the capital city. Future studies should consider the views of media workers located outside the capital. Audience studies to ascertain public understanding about vaccine trials will help to determine if the subsequent public education efforts made a difference in public understanding of vaccine trials. 

Based on this discussion, some recommendations are in order. It is important that trial implementers of sensitive (and possibly controversial) vaccine trials engage the media after trial approval, but prior to the start of trials, to explain the trial processes. That way, implementers can address concerns that the media may have. It also provides the media with accurate information for dissemination and spokespersons that the media can rely on. Such collaboration with the media will have the added benefit of educating the media about trials broadly and vaccine trials specifically. 

Secondly, there is always the potential for trial unacceptance in future investigations. To address such concerns, volunteers from previous trials can be recruited to help in public sensitization activities prior to the start of trials. These “advocates” are more likely to be believed and accepted than the trial implementers. It is high time local research review processes for such trials include a requirement for local sensitization as part of the approval process. Using the biomedical explanation of the public good is unlikely to be accepted in the current risk society milieu. It is critical that trial implementers get the buy-in or at least inform community leaders and chiefs. Although it may be viewed as another “roadblock,” it will ensure that important trials are not curtailed by perceptions of surreptitious activities. Sensitization activities may not always be required. For instance, previous malaria vaccine trials did not face resistance, but the 2019 malaria vaccine deployment in Ghana faced controversies although it was not curtailed [41].

All in all, I argue for sensitivity to contextual, historical, and cultural factors when conducting vaccine trials among groups that are viewed as “resistant” to such interventions. It is important to include people of diverse backgrounds in vaccine trials to test their efficacy for broad use. The study shows that different frames of reference are used by diverse groups during risky situations. The context of a risk society (the fear of Ebola) provides a conducive environment for moral panic to develop, particularly when the legacy media is perceived as a critical and credible source of information. An effect of the suspended vaccine trials was a weakening of the journalist–scientist relationship. Indeed, some scientists later chastised the media for how they handled the issue. This could lead to challenges with communication during other disease outbreaks or health risk situations.

## Data Availability

The data presented in this study for the content analysis is publicly available on the websites of the digital platforms. Data for interviews is available on request from the corresponding author.

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
