# Peer review of "Botched Ebola Vaccine Trials in Ghana: An Analysis of Discourses in the Media"

_vaccines, 2021, doi:10.3390/vaccines9020177_

Round 1
Reviewer 1 Report
The study conducted and described by Thompson is highly interesting and relevant to the current world-wide pandemic. The information presented will likely help to guide the public relations of future vaccine studies in Ghana.
Minor comments:
-It would be helpful if the statements from interviewees were more obviously separated from the author's own text and delimited from each other. Perhaps with quotation marks or using increased indentation. Some sections feel jarring as the reader must infer the source (author or interviewee) from the style and pronouns.
-While the results are interesting, it is not obvious what was contributed by the procedures done under section 3.3 Coding and Analysis. It could be useful to mention it in the relevant sections when these results are used. Would it be possible (without violating privacy) to share, perhaps as supplementary materials, some of the more "raw" results from the analyses performed in that section?
-While the author is not expected to solve the issues described in the manuscript, it would be useful to have some more concrete suggestions for future vaccine studies, if the author can think of some. For example, suggesting that companies approach journalists directly as soon as a trial is approved so that accurate information can be distributed widely and early.
Author Response
Thank you for the feedback. I have responded to the comments raised below as follows:
The study conducted and described by Thompson is highly interesting and relevant to the current world-wide pandemic. The information presented will likely help to guide the public relations of future vaccine studies in Ghana.
Thank you.
Minor comments:
-It would be helpful if the statements from interviewees were more obviously separated from the author's own text and delimited from each other. Perhaps with quotation marks or using increased indentation. Some sections feel jarring as the reader must infer the source (author or interviewee) from the style and pronouns.
Thank you. It seems setting the manuscript in the journal formatting removed all previous formatting. I have set off all quotations that are more than 40 words in block quotes. However, quotations that are less than 40 words are presented in quotation marks within the paragraph. Please see the findings section for these changes.
-While the results are interesting, it is not obvious what was contributed by the procedures done under section 3.3 Coding and Analysis. It could be useful to mention it in the relevant sections when these results are used. Would it be possible (without violating privacy) to share, perhaps as supplementary materials, some of the more "raw" results from the analyses performed in that section?
Thank you. I have included an appendix that shows the three levels of coding and analysis employed with sample content. This is presented as Appendix 1 titled “sample coding and analysis process.”
-While the author is not expected to solve the issues described in the manuscript, it would be useful to have some more concrete suggestions for future vaccine studies, if the author can think of some. For example, suggesting that companies approach journalists directly as soon as a trial is approved so that accurate information can be distributed widely and early.
Thank you. In the discussion section on page 13, I have included a paragraph on recommendations. Here, I suggest the need for trial implementers to include media engagement processes after review board approval but prior to recruitment in order to build media buy-in and help in public education. This is not a general suggestion but should be only used in situations where the trials are likely to cause anxiety among the general population. I also suggest that local approval processes should include engaging or at least informing local authorities in the trial sites about trials. I also suggest the use of previous volunteers as “advocates” especially in situations where potential "unacceptance" of trials are likely to occur. These are found in the third paragraph of page 13.
Reviewer 2 Report
The last decade has shown how serious a global public health threat posed by filoviruses can be. The most devastating epidemic, which took place in West Africa from 2013 to 2016 and was caused by Ebola virus, and also the most recent outbreaks in the Democratic Republic of Congo, highlight an urgent need for the development of medical countermeasures against filovirus infections. The ultimate solution to control filovirus-induced disease is a safe and effective vaccine. However, it normally takes large-scale trials in human population to gain solid and reliable data on vaccine safety and immunogenicity, before it becomes widely available for people as a measure of infection prophylaxis. The results of these trials are not only determined by scientific efforts invested into development of experimental vaccine, but are strongly dependent on whether such trial initiatives are benevolently met by the society, an aspect which does not seem to be given enough attention in the current literature. From that perspective, the study of Esi E. Thompson «Botched Ebola Vaccine Trials in Ghana: An Analysis of Discourses in The Media» serves as a nice example of that kind of analysis, revealing the reasons of suspension of Ebola vaccine trials in Ghana in 2015. It clearly demonstrates that, no matter how encouraging the results of preclinical vaccine studies might be, the trial is doomed to fail when it is seen as harmful and unethical by public. The study shows how discourses in the mass media can provoke moral panic and risk society and push the decision-makers to halt very important vaccine trials. The social mechanisms of rejection need to be thoroughly considered when planning such trials, and require future investigations.
One comment I have which can improve manuscript is that author may add some practical suggestions into the Discussion on how to deal with potential unacceptance of vaccine trials by public during the future outbreaks.
In sum, the study is solid, timely, well-written, the results are convincing, and the conclusions are fully supported with the presented data. The paper by Esi E. Thompson entitled «Botched Ebola Vaccine Trials in Ghana: An Analysis of Discourses in The Media» is suitable for publication in Vaccines journal.
Author Response
Thank you for the feedback. I have responded to the comments raised below:
Reviewer 2 made the following observations:
The last decade has shown how serious a global public health threat posed by filoviruses can be. The most devastating epidemic, which took place in West Africa from 2013 to 2016 and was caused by Ebola virus, and also the most recent outbreaks in the Democratic Republic of Congo, highlight an urgent need for the development of medical countermeasures against filovirus infections. The ultimate solution to control filovirus-induced disease is a safe and effective vaccine. However, it normally takes large-scale trials in human population to gain solid and reliable data on vaccine safety and immunogenicity, before it becomes widely available for people as a measure of infection prophylaxis. The results of these trials are not only determined by scientific efforts invested into development of experimental vaccine, but are strongly dependent on whether such trial initiatives are benevolently met by the society, an aspect which does not seem to be given enough attention in the current literature. From that perspective, the study of Esi E. Thompson «Botched Ebola Vaccine Trials in Ghana: An Analysis of Discourses in The Media» serves as a nice example of that kind of analysis, revealing the reasons of suspension of Ebola vaccine trials in Ghana in 2015. It clearly demonstrates that, no matter how encouraging the results of preclinical vaccine studies might be, the trial is doomed to fail when it is seen as harmful and unethical by public. The study shows how discourses in the mass media can provoke moral panic and risk society and push the decision-makers to halt very important vaccine trials. The social mechanisms of rejection need to be thoroughly considered when planning such trials, and require future investigations.
Thank you.
One comment I have which can improve manuscript is that author may add some practical suggestions into the Discussion on how to deal with potential unacceptance of vaccine trials by public during the future outbreaks.
Thank you. In the discussion section on page 13, I have included a paragraph on recommendations. Here, I suggest the need for trial implementers to include media engagement processes after review board approval but prior to recruitment in order to build media buy-in and help in public education. This is not a general suggestion but should be only used in situations where the trials are likely to cause anxiety among the general population. I also suggest that local approval processes should include engaging or at least informing local authorities in the trial sites about trials. I also suggest the use of previous volunteers as “advocates” especially in situations where potential unacceptance of trials are likely to occur. These are found in the third paragraph of page 13.
In sum, the study is solid, timely, well-written, the results are convincing, and the conclusions are fully supported with the presented data. The paper by Esi E. Thompson entitled «Botched Ebola Vaccine Trials in Ghana: An Analysis of Discourses in The Media» is suitable for publication in Vaccines journal.
Thank you.
Reviewer 3 Report
First, I have to admit that this is not a biomedical paper for which I would feel confident to provide a decent review. This is a sociological paper. The language is quite plane, and I’m not sure whether it is accepted in research papers in sociology – this is not my area, but from my prospective this manuscript does not sound as a research or review paper at all. Even sections (e.g., “2. Literature review”, “Findings”) does not sound suitable for a biomedical publication. The author several times mentioned that he performed qualitative analysis. However, even qualitative analysis would require some quantitative elements instead of saying “some expressed support…”, “other journalists chastised…”, “other interviewees suggested…”, “some medical workers made claim…”, etc. Reading this, I do not see which opinions, influences, outcomes prevailed, and how the situation was developing in the way that it did. It would be helpful if author shows numbers or proportions of specific (positive and negative) publications, underlying drivers and mechanisms of the processes related to the vaccine trials, levels of authority of the individuals expressing different opinions that influenced the decisions made. It would be also useful to see the analysis of official publications from the Ministry of Health, and their public health outreach. I do not understand why FDA is quoted so frequently compared to Ghanaian health authorities. I do not see this paper as anything different from another “media” assay, including author’s discussion and concluding remarks. I do not value this manuscript as a scientific paper and cannot recommend it for publication in “Vaccines”.
Author Response
Reviewer 3 made the following observations
First, I have to admit that this is not a biomedical paper for which I would feel confident to provide a decent review. This is a sociological paper. The language is quite plane, and I’m not sure whether it is accepted in research papers in sociology – this is not my area, but from my prospective this manuscript does not sound as a research or review paper at all.
Thank you. Yes, this is a sociological paper and follows the conventions of such manuscripts.
This manuscript is not a biomedical paper and I identify the paper as using a sociological lens. Although Vaccines is a biomedical journal, the call for papers for the special issue “Vaccines for Ebola Virus and Related Diseases," requested contributions on all aspects of the “design, development, testing and administration of current and future vaccines for Ebola virus and related viruses.” The call further invited contributions addressing “Ebola vaccine policy and social issues relating to accessibility of vaccines, Ebola vaccine use during humanitarian crises, epidemiological methods for assessing Ebola vaccine efficacy and safety, preventive and reactive uses of the vaccine, vaccine distribution, vaccination of special groups including pregnant women, infants and children and persons with simultaneous co-infections, vaccines and sexual transmission, Ebola vaccine hesitancy, and other important topics relating to these life-saving vaccines.” The call was not restrictive about methods or approaches. This manuscript responds to the call using qualitative methods to address an area that is often neglected in research about vaccine, but which can have a strong impact on the vaccine development process. Vaccine development for the coronavirus has brought to the fore the need to include diverse population segments in trials as well as questions about why some population segments are hesitant to participate in trials. Until we understand these issues, calls for including diverse populations in trials will remain calls. Different structural and systemic issues exist around vaccine hesitancy that quantitative methods cannot unearth (see Lindloff and Taylor, 2011). I adopted methods that would best address the issues at stake by capturing the discourses behind the suspension of the trials (Gillespie, 2001; Lindloff & Taylor, 2011).
Even sections (e.g., “2. Literature review”, “Findings”) does not sound suitable for a biomedical publication. The author several times mentioned that he performed qualitative analysis.
I had a section on literature review because I adopted two theories—moral panic and risk society—and it is important to help readers understand what these theories mean and how they have been used in the study.
However, even qualitative analysis would require some quantitative elements instead of saying “some expressed support…”, “other journalists chastised…”, “other interviewees suggested…”, “some medical workers made claim…”, etc. Reading this, I do not see which opinions, influences, outcomes prevailed, and how the situation was developing in the way that it did. It would be helpful if author shows numbers or proportions of specific (positive and negative) publications, underlying drivers and mechanisms of the processes related to the vaccine trials, levels of authority of the individuals expressing different opinions that influenced the decisions made.
Thank you. Qualitative methods are legitimate methods for undertaking empirical research and provide insights that are critical for health generally.
Using an interpretivist lens does not privilege one view over another or one outcome over another. In addition, the use of non-probabilistic samples makes it meaningless to privilege a view over another (Zoller & Klien, 2008). Imposing numbers (a quantitative reductionist approach) contrasts with the interpretivist epistemological (in the constructionist tradition) approach used in this study (Lindloff & Taylor, 2011). I did include the number of interviewees, (18 across legacy media) 21 news stories and 245 user comments. Furthermore, as I stated in the paper, most stories provided information from both the trial opposers and trial supporters in an effort to be balanced (page 8 paragraph 2). Counting the number of positive and negative stories would not be insightful in this case. In this manuscript, I use risk related theories to understand an event that occurred based on discourses in the media. This process allows one to understand the different opinions and positions taken and held. As I explain in the manuscript, two main positions were taken: trial supporters and trial opposers irrespective of whether it was the GAAS, members of parliament, or commenters on social media. Both the WHO and the Minister of Health spoke in favour of the trials, but it did not make a difference in the final outcome.
It would be also useful to see the analysis of official publications from the Ministry of Health, and their public health outreach.
Thank you. The focus of the study was on discourses in the media and data were therefore gathered from media sources. Analysis of official publications may prove insightful for future studies, but for the current study, I analyzed discourses in the media where the voices of various stakeholders (formal, local, political, professional, international etc) were present.
I do not understand why FDA is quoted so frequently compared to Ghanaian health authorities.
Thank you. The FDA approves vaccine trials in Ghana and also oversees trial procedures. The Ghana Health service is not involved in that process. Therefore, any issues about vaccine trials fall squarely within the FDA and they are likely to be referenced. Secondly, as explained in section 1.1, the Ebola vaccine trials had not been approved when the issue broke out. But the FDA subsequently approved them and held a press conference to announce the approval. Therefore, as the government agency mandated to approve trials, they would certainly be targeted in much communication and discourse about the vaccine trials.
I do not see this paper as anything different from another “media” assay, including author’s discussion and concluding remarks. I do not value this manuscript as a scientific paper and cannot recommend it for publication in “Vaccines”.
Thank you, but, the basis of this recommendation is unfortunate and does not address the manuscript on its terms. Using a quantitative biomedical lens to review a qualitative sociological manuscript does not do justice to the manuscript. Not only is this unfair, it devalues other empirical methods of research.